# An Overview on Down-Looking UAV-Based GPR Systems

**Carlo Noviello** [1,*] , **Gianluca Gennarelli** [1] , **Giuseppe Esposito** [1,2] , **Giovanni Ludeno** [1] , **Giancarmine Fasano** [2] ,
**Luigi Capozzoli** [3] , **Francesco Soldovieri** [1] **and Ilaria Catapano** [1]

[1] Institute for Electromagnetic Sensing of the Environment (IREA), National Research Council (CNR),
80124 Napoli, Italy; gennarelli.g@irea.cnr.it (G.G.); esposito.g@irea.cnr.it (G.E.); ludeno.g@irea.cnr.it (G.L.);
soldovieri.f@irea.cnr.it (F.S.); catapano.i@irea.cnr.it (I.C.)
[2] Department of Industrial Engineering (DII), University of Naples "Federico II", Via Claudio 21,
80124 Napoli, Italy; g.fasano@unina.it
[3] Institute of Methodologies for Environmental Analysis, National Research Council,
C.da S. Loja-Zona Industriale, 85050 Potenza, Italy; luigi.capozzoli@imaa.cnr.it
* Correspondence: noviello.c@irea.cnr.it; Tel.: +39-0817620656

**Abstract:** Radar imaging from unmanned aerial vehicles (UAVs) is a dynamic research topic attracting huge interest due to its practical fallouts. In this context, this article provides a comprehensive review of the current state of the art and challenges related to UAV-based ground-penetrating radar (GPR) imaging systems. First, a description of the available prototypes is provided in terms of radar technology, UAV platforms, and navigation control devices. Afterward, the paper addresses the main issues affecting the performance of UAV-based GPR imaging systems. such as the control of the UAV platform during the flight to collect high-quality data, the necessity to provide accurate platform position information in terms of probing wavelength, and the mitigation of clutter and other electromagnetic disturbances. A description of the major applicative areas for UAV GPR systems is reported with the aim to show their potential. Furthermore, the main signal-processing approaches currently adopted are detailed and two experimental tests are also reported to prove the actual imaging capabilities. Finally, open challenges and future perspectives regarding this promising technology are discussed.

**Keywords:** radar imaging and signal processing; ground-penetrating radar; unmanned aerial vehicle; linear inverse scattering; global positioning systems

## 1. Introduction

Modern unmanned aerial vehicles (UAVs) represent a very attractive technology due to their simple use, low maintenance costs, and high operational flexibility. UAV platforms can take off and land from very small areas and can move in all directions [1]. These peculiar features enable their use almost at any location and under different flight modes. UAVs are powerful tools in several scientific fields where they are often integrated with passive sensors (such as optical (e.g., visible or infrared) cameras) and/or active sensors (such as light detection and ranging (LIDAR) and radar).

Among the available sensing technologies, radar systems can record data in all-weather and day/night conditions and, thanks to the ability of electromagnetic waves to penetrate visually opaque materials, they can image buried/hidden objects. For this reason, the ground-penetrating radar (GPR) [2] is gaining increasing attention as a complementary remote sensing instrument onboard UAV platforms, and the scientific community, as well as various industries, is attempting to develop innovative and effective UAV-based GPR systems [3–23]. The combination of UAV and GPR technology can lead to the creation of innovative imaging systems, which could be exploited in many applicative contexts, such as landmine detection [3,5–7,9,10,15,17], glaciology [8,19,23], search and rescue [8], agriculture [16], environmental monitoring [4,11,18], and cultural heritage [12].

Over the last decade, many UAV-based radar imaging systems have been developed based on different radar technologies, UAV platforms, and payload solutions [3–26]. UAV GPR systems may be broadly classified into two groups. The first group includes innovative prototypes specially assembled and integrated for operating onboard UAV platforms. The second group regards systems based on commercial GPR devices, designed for operating in standard mode (i.e., with radar antenna almost in contact with the investigated surface), and mounted on board a UAV platform. As discussed in [27], UAV-based GPRs can operate according to different observation modalities: forward-looking, down-looking (sounder), and side-looking. Each measurement configuration has its pros and cons. Forward- and side-looking configurations seem to be more effective in detecting shallow targets since the signal reflection from the air–soil interface is minimized; on the other hand, the down-looking configuration is more suitable to detect deeper targets due to the improved dynamic range.

Flight dynamics is a major problem related to UAV-based GPR imaging typically resulting in non-rectilinear flight trajectories characterized by a variable speed and height above the air–soil interface. As a result, the quality of radar data can be impaired, leading to defocused images of the scene under investigation [28]. The effects of motion errors can be mitigated, at least partially, by resorting to proper hardware and signal-processing solutions. The first one is based on the use of sophisticated devices which can estimate (in real-time or post-processing) the UAV position with high accuracy [3,5,7–9,12,15,17,19]. Other solutions exploit advanced signal-processing procedures to estimate and compensate for the altitude variations of the UAV platform directly from the radar data [11]. Other relevant issues for UAV-based GPRs are their reduced penetration capability compared to conventional ground-based systems, the additional clutter due to lateral reflections from surface targets, and electromagnetic disturbances present in the environment.

This article deals with an overview on UAV-borne GPR imaging based on the experience gained by the authors on the topic. In this frame, note that [27] mostly deals with UAV GPR systems for the detection of landmines and improvised explosive devices (IEDs) operating in forward-looking, side-looking, and down-looking modes. This manuscript focuses on state-of-the-art UAV-borne GPR systems operating in down-looking modality from a chronological perspective, starting from the first conceptual studies [3] up to the recent commercial systems [20–22]. The manuscript also addresses the technological evolution of these systems and the challenges that have been faced and solved by exploiting existing solutions or developing new hardware and software strategies. A comprehensive review of each system documented in the literature is presented in terms of radar technology, UAV platforms, and payload electronic devices. This manuscript, differently from [27], provides also a detailed description of the main issues affecting UAV-based GPR imaging and the technological solutions implemented for mitigating them.

The paper focuses also on signal-processing strategies implemented in UAV-based GPR prototypes. These techniques are herein classified as standard GPR processing and advanced imaging algorithms. Standard GPR processing involves time/frequency procedures, which can compensate the UAV trajectory deviations, attenuate the signal, and mitigate the clutter. Advanced imaging algorithms typically follow the standard processing and aim at improving the interpretability of the results. The most widely adopted strategies are the Synthetic Aperture Radar (SAR) back-projection method [6,9,14] or *f-k* migration [29,30]. Another relevant focusing strategy worth mentioning is the microwave tomography (MWT) approach based on the solution of a linear inverse scattering problem [31].

The paper is organized as follows. Section 2 describes the UAV GPR prototypes in terms of technological solutions, i.e., radar technology, UAV platform, and the onboard electronic devices. Section 3 addresses the main issues affecting the performance of UAV-based radar imaging systems and the main strategies used to mitigate them. Section 4 provides an overview of the applicative contexts where the prototypes are employed. Section 5 deals with the main data-processing approaches implemented by the actual prototypes. Section 6 reports two experiments involving surface and subsurface imaging

to evaluate the achievable imaging performance and demonstrate the potential of these systems. Finally, the manuscript ends with a discussion about the open challenges and the future perspectives of UAV-borne radar imaging technology.

## 2. State-of-the-Art Down-Looking UAV-Based GPR Systems

In recent years, the scientific community has expressed considerable interest in down-looking UAV-based GPR systems thanks to the improvements in UAV functionalities, such as autonomous flight capability, multiple sensors integration, and innovative UAV platforms, together with a significant reduction in costs and maintenance operations. However, being light and compact platforms, mini UAVs (weight of <25 kg) and micro UAVs (weight of <1 kg) [32] suffer from several constraints which limit the maximum payload mass. Indeed, some trade-offs must be found during the design stage in terms of radar technology, GPR antennas, the number and type of navigation instruments, batteries, etc. Despite that, due to the continuous technological advances, the world of UAV-GPR systems is evolving and there is still room for improvement.

This section provides a chronological survey of the main UAV-based GPR systems, which are here denoted as System 1, System 2, System 3, etc., following the order by which they are introduced.

The first imaging radar specifically developed for a UAV platform was the airborne radar for three-dimensional imaging and nadir observation (ARTINO) (System 1) [3], designed at Fraunhofer Institute for High-Frequency Physics and Radar Techniques FHR. System 1 combines a real aperture, made by a linear array of nadir-pointing antennas (sounder mode), and SAR technology, which is spanned by the moving UAV platform. The radar front-end uses a frequency-modulated continuous wave (FMCW) technique and operates in the Ka band. This technology reduces the radar power consumption and the size of the antennas. Even though System 1 has no penetration capabilities, it is mentioned here because it is the first UAV-based radar system operating in the same observation modality of a GPR. System 1 is characterized by a multi-static antenna array distributed along the wing of a small aircraft. The multi-static array is exploited to scan the antenna beam in the across-track direction and to resolve the left–right ambiguity of a sounder system. Upon flying at a reference altitude of 200 m, ARTINO can illuminate a ground strip width of approximately 230 m and produce a 3D image of the scene with a cubic resolutions cell of about 20 cm × 20 cm × 20 cm. The navigation unit of System 1 is composed of a differential global positioning system (DGPS) integrated with acceleration sensors and gyroscopes for each axis. The navigation unit is also capable of tracking the antenna orientation during the data recording stage by measuring the oscillations and detecting the wings with the help of a laser/CCD unit. The navigation unit is designed to provide the status and position of the platform to an autonomous flight system or to an operator to correct the platform trajectory in real time during the data collection stage.

The second UAV radar imaging system (System 2) mentioned here is a simple and compact prototype developed at the University of Texas in 2016 by mounting a commercial radar system on board a micro-UAV platform coupled with an optical camera [4]. The radar module is the ultra-wideband Pulson P410 sensor operating in the frequency range of 3.1–5.3 GHz, which achieves a 10 cm range resolution. System 2 has two helix antennas (1 Tx and 1 Rx) with a spacing less than half wavelength between them so operating in monostatic mode. The radar and the camera are controlled onboard via a Raspberry Pi 2 computer, connected with a ground-based laptop using a Wi-Fi connection. The entire system (including batteries and cables) weighs less than 300 g and is mounted on a very light and compact UAV DJI Phantom 2 drone. System 2 is tested by performing flights over objects located 2–3 m above the ground (e.g., trees, cars and people), proving the system detection capability.

The third UAV GPR system (System 3) was developed in 2017 by a research group at the Pontificia Universidad Javeriana in Colombia [5]. System 3 is a custom-designed lightweight GPR based on the innovative software-defined radio (SDR) technology. The

GPR is composed by the programmable electronic board USRP B210, a Tx-Rx antenna pair, and SMA connectors. The SDR module transmits a pulse with a 56 MHz bandwidth, modulated by a carrier signal at 2 GHz in the form of a raised cosine filter (RCF). System 3 is equipped with two Vivaldi Antipodal antennas, mounted in bistatic configuration with an inclination of 45°, and specially designed and fabricated for radar applications. The antennas are lightweight with a thickness of 1 mm and are characterized by a symmetrical radiation pattern in the frequency range of 1.5–9.0 GHz. The entire GPR module is very compact and its weight is about 330 gr. The navigation control unit of System 3 consists of high-level and low-level processors, a ZigBee communication module, an inertial measurement unit (IMU), a GPS, and a laser sensor (LIDAR-lite model LL-905). The navigation control unit allows a very accurate real-time control of the UAV flight path, enabling the drone to fly steadily at a constant altitude (e.g., 50 cm) above the ground.

System 4 is an ultra-wide-band (UWB) GPR system developed in 2018 by the University of Ulm in collaboration with Endress and Hauser GmbH company and the University of Applied Sciences and Arts of Northwestern in Switzerland. System 4 is a compact and flexible UAV-borne radar system specially designed for landmine detection [6]. It is composed by a DJI Matrice 600 Pro UAV platform, a GPR system, a navigation control unit, and a data logger. The GPR front-end is based on the FMCW technology and works in the frequency range of 1–4 GHz. The GPR is equipped with two lightweight 3D-printed horn antennas (1 Tx and 1 Rx), which can operate both in standard down-looking and side-looking modes. The navigation control unit is composed of several devices: a 26 GHz radar and a LIDAR altimeter which can measure the UAV flight altitude, a real-time kinematic (RTK) global navigation satellite system (GNSS) system, three GPS receivers, and three IMUs. The RTK module is interconnected with the UAV autopilot in order to follow the planned path and correct the position deviations in real time. GPR and navigation devices are connected with the data logger, which stores the data and provides a time stamp to each measurement, thus ensuring the data synchronization. System 4 was recently equipped with two additional horn antennas to implement a single-pass interferometric radar configuration [7].

System 5 is an ultra-wide-band snow sounder (UWiBaSS) GPR developed by Norut Northern Research Institute (Norway) in 2018 for snowpack surveying [8]. The radar module is the M-Sequence UWB sensor developed by the German company ILMSENS (https://www.uwb-shop.com/, accessed on 9 June 2022) with a weight of 4 kg. This sensor can perform two parallel acquisitions by exploiting two receiving channels controlled by an onboard computer with customized software. The GPR is equipped with one transmitting spiral antenna and two receiving Vivaldi antennas mounted with 90 degrees offset one to each other. This experimental set-up allows the detection of phase differences between the target radar cross-sections. The measurement bandwidth is equal to 5.05 GHz and covers the frequency range of 0.95–6 GHz. The radar module can achieve a 5 cm range resolution in the air with an unambiguous range of 5.75 m. System 5 exploits the Kraken octocopter as a UAV platform, which allows a maximum takeoff weight of 20 kg; the weight of the empty system is 8.5 kg, while the maximum payload mass is about 11.5 kg. Furthermore, System 5 includes a LIDAR, a GPS receiver, and a RTK module as auxiliary navigation devices. The LIDAR is mounted on one of the eight arms of the UAV and accurately tracks the range distance from the ground; the GPS receiver and the RTK module are exploited to accurately control the UAV position during the flight.

Two similar UAV-borne GPR prototypes (System 6 and System 7) were developed for the detection of buried objects by research groups at the University of Oviedo, University of Madrid, and University of Vigo, in 2018 [9] and 2019 [10], respectively. A picture of System 6 is provided in Figure 1. Systems 6 and 7 use the same UAV platform and navigation control unit, but they differ by the radar modules and antennas. Both systems exploit a custom version of DJI Spreading Wings S1000 as a UAV platform. This last can support a maximum payload of 6 kg and can fly for 15 min at full load. The navigation control unit is composed of an IMU, a LIDAR altimeter, a barometer, and a receiver GNSS

station connected with a RTK module by using a wireless data link. This architecture is used to achieve UAV positioning accuracy at the centimeter scale. The navigation control, the UAV autopilot, and the radar sensor are integrated by means of a data logger computer to synchronize the radar data with the position information. The radar sensor employed in System 6 is the Pulson P410 UWB module, already described for System 3. The radar is equipped with two customized helix antennas—one transmitting in Right-Handed Circular Polarization (RHCP) and the other one receiving in Left-Handed Circular Polarization (LHCP). Conversely, System 7 uses the M-Sequence UWB sensor exploited by System 5 as the radar. In System 7, two different antennas (Tx and Rx) were considered. In the first stage, two UWB Vivaldi antennas working in the frequency range of 600 MHz–6 GHz are tested [10]. More recently, System 7 was equipped with two log-periodic antennas [25]. A dedicated structure was designed and manufactured with 3D printing technology to mount the radar on board the UAV and isolate the antennas.

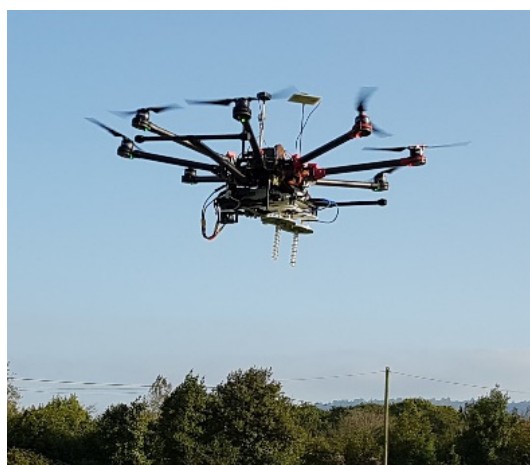

**Figure 1.** Picture of UAV-GPR System 6. Courtesy of University of Oviedo, Gijón, Spain.

In 2018, a UAV-borne GPR prototype (System 8) was proposed by the Institute for the Electromagnetic Sensing of the Environment of the Italian Research Council (IREA-CNR) in collaboration with the Department of Industrial Engineering (DII) of University of Naples "Federico II" [11]. The system shown in Figure 2 uses the Pulson P440 radar module, which is the updated version of the Pulson P410 used by System 2 and System 6. The radar is equipped with two log-periodic PCB antennas (Ramsey LPY26)—one transmitting and the other receiving, operating in a sounder mode. These antennas have a broad radiation pattern and are closely spaced from each other in terms of radiated wavelength, such that a monostatic configuration is implemented. The UAV platform is the DJI F550 hexacopter, which allows a maximum payload mass of 1 kg. The UAV is controlled through a DJI Naza M-Lite autopilot, which is located in the hexacopter center of mass and complemented by a remote GPS/compass installed on a mast. The onboard payload for the radar survey includes the radar system, a compact Odroid Linux-based computer, and an auxiliary GNSS receiver (Ublox LEA-6T). The GPS and radar data acquisitions are controlled by custom software running on the Odroid, which is necessary to perform the synchronization between radar and GPS data. Recently, System 8 was also equipped with a ground-based GNSS receiver [13,14], allowing the implementation of the post-processing kinematic (PPK) carrier-based differential GPS (CDGPS) technique. This upgrade was crucial to achieve positioning information at the centimeter scale with an improvement in the imaging performance [13].

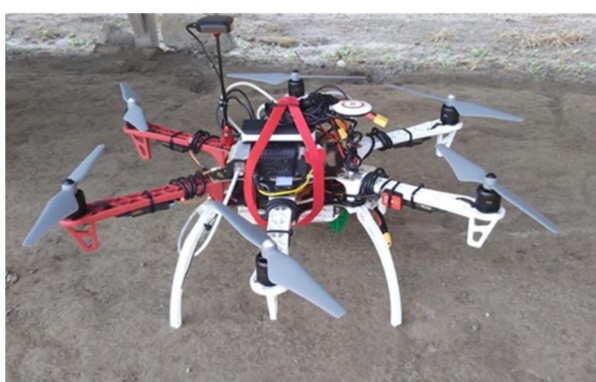

**Figure 2.** Picture of UAV GPR System 8, Courtesy of Department of Industrial Engineering (DII) of University of Naples, "Federico II".

System 9 (see Figure 3) is the evolution of the P3M-SAR system developed in collaboration between the German Aerospace Center (DLR) and the Karlsruhe Institute of Technology in 2018 [15]. System 9 is made of three main parts: (1) the UAV platform, (2) the sensor payload, (3) and the ground control station managing UAV flight data and sensor data. System 9 uses a customized version of DJI® Matrice 600 pro drone. It can fly for about 15 min and carries a payload of 6 kg. The UAV payload is composed of the radar module, the radar antennas, a high-resolution optical camera, and a navigation measurement unit. The radar module is based on the FMCW technology and operates in the frequency interval 500 MHz–3 GHz, which represents a good compromise between the penetration capabilities and range resolution. The size of the sensor module is 10 cm × 10 cm × 5 cm and its mean output power is 250 mW. The radar has two Vivaldi antennas operating in bistatic configuration with a beamwidth of about 70 degrees in the azimuth plane. System 9 is able to perform a SAR stripmap and more sophisticated flight trajectories (i.e., circular and helical). In the SAR stripmap mode, System 9 provides a 2D image with a 6 cm × 6 cm range and azimuth resolutions, respectively, while a 3D radar image with resolutions of about 4 cm × 4 cm × 4 cm can be achieved in the circular mode. The navigation unit of System 9 is equipped with a RTK GPS module and an IMU. System 9 is designed to send all data (radar, camera, and navigation data) to a ground station by exploiting a 433 MHz data link. In this way, the system processor should be able to receive the SAR image and a 3D model of the ground surface in near real time. However, at present, a part of the data is stored onboard and another part is transferred to the ground station, where the image processing is carried out offline.

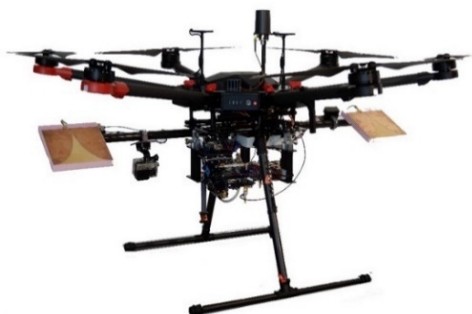

**Figure 3.** Picture of UAV GPR System 9. Courtesy of Deutsches Zentrum für Luft- und Raumfahrt (DLR), Germany.

In 2019, Wu et al. [16] proposed a UAV-borne GPR prototype (System 10) based on a vector analyzer equipped with a specially designed antenna. The vector analyzer is the Planar R60 model, produced by Copper Mountain Technologies, with dimensions of 130 mm × 65 mm × 28 mm and a weight of about 350 g. The VNA is configured to work as a stepped frequency continuous wave (SFCW) radar in the frequency range of

1 MHz–6 GHz. The radar operates in the monostatic mode by using a hybrid horn dipole antenna covering the typical GPR frequency range, i.e., 250–2800 MHz. The antenna was built on the continuation of the horn aperture to cover lower frequencies without significantly increasing the antenna dimension and weight. The UAV platform is the X8 model made of eight motors and four arms, and has a maximum payload of 7 kg. The UAV platform is managed by a microcomputer, remotely controlled by a smartphone. The whole radar system weighs 1.5 kg. The imaging capabilities of System 10 are assessed by retrieving the soil moisture map of the surveyed area.

The UAV-based GPR system specifically designed for landmine detection in 2020 at University of Maribor, Slovenia, (System 11) [17] is also worth mentioning. As shown in Figure 4, a lightweight and low-power GPR is specially designed to be mounted onboard a mini UAV. The radar system is based on the SFCW technology, operates in quasi-monostatic configuration (distance between antennas less than half wavelength), and can be used in both down-looking and forward-looking modes. Two hybrid Vivaldi-Horn (VH) antennas, with a thickness of 0.5 mm, are fabricated with a thin copper plate and fed over an SMA connector. The antenna has a size of 95 mm × 225 mm × 180 mm and a weight of 240 g, and is suitable for middle to micro UAVs. The frequency band of antennas range from 550 MHz up to 2.7 GHz, and the maximum available bandwidth is 2.15 GHz. The GPR total weight is about 780 g. A complete description of the electronic components of the radar front-end is available in [17]. The UAV drone platform DJI Matrice 600 Pro is exploited to assess the GPR performance. This platform has been previously described by other systems and is characterized by a flight autonomy of 18 min with a maximum payload of 5.5 kg. The GPR is connected to the flight controller with an open-source software development kit (SDK) provided by the UAV manufacturer. The software exploits digital inputs to trigger the scanning remotely. The navigation control units consist of a GNSS receiver and an IMU that provides velocity, attitude, and other information. All navigation data are stored onboard during the survey using a data logger.

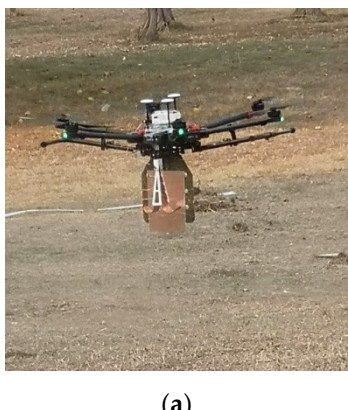 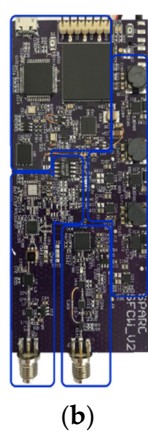 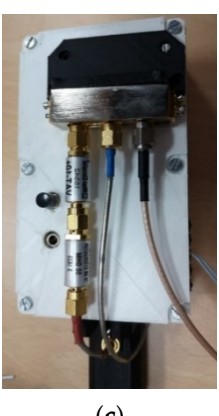

(**a**) (**b**) (**c**)

**Figure 4.** Picture of UAV GPR System 11: UAV platform (**a**); transceiver module (**b**); cable interconnections (**c**). Courtesy of University of Maribor, Maribor, Slovenia.

Recently, another UAV GPR solution (System 12) has been proposed in [18]. System 12 (see Figure 5) consists of the COBRA Plug-in SE-150 monostatic antenna [18] mounted on the DJI Matrice 600 Pro platform. The radar module operates at 120 MHz with 240 MHz bandwidth. The radar system helps to explore the subsoil with a nominal vertical resolution up to 27 cm and a penetration depth up to 40 m in a medium with a relative dielectric permittivity equal to 5. The UAV payload is composed of an IMU and a GNSS receiver for the management and control of the UAV flight trajectory. A data logger system is also mounted onboard the drone, allowing GPR data logging and instrument control from the ground station. The drone is also equipped with an automatic terrain-follow sensor, which can estimate the sensor to ground distance with high precision. The UAV platform is the

standard version of the DJI Matrice 600 Pro; therefore, it nominally provides the same performance of the DJI Matrice Pro platform previously described.

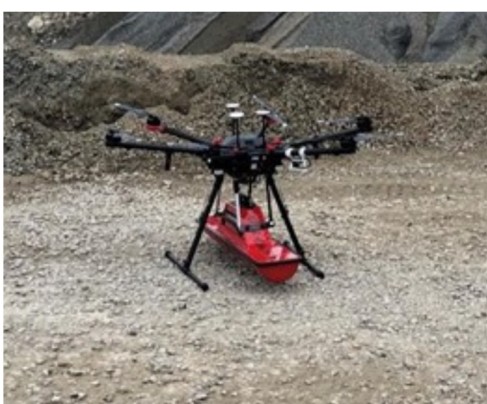

**Figure 5.** Picture of UAV GPR System 12. Courtesy of Dyrecta Lab., Conversano, Italy.

Different research activities recently conducted by SPH Engineering Company on UAV-based GPR are also worth considering [19]. SPH engineering developed a UAV GPR system (System 13), which is capable of exploiting three different commercial GPR sensors: Cobra Plug-In GPR [20] and Cobra CBD GPR [21] (both produced by Radarteam, Boden, Sweden), and the radar systems' Zond-12e (manufactured by Radar Systems Inc., Riga, Latvia) [22]. The Cobra Plug-In GPR is a radar kit composed of three antennas: Model SE-40, operating in the frequency range of 15–105 MHz; Model SE-70, operating in the range of 20–140 MHz; and Model SE-150, working in the range of 20–280 MHz. Cobra CBD is a multi-frequency radar operating at 200/400/800 MHz with a variable bandwidth in the range of 50–1400 MHz. The Zond-12e is a radar module characterized by two shielded antennas operating in the frequency range of 200–900 MHz. The System 13 exploits two UAV platforms: DJI® Matrice M600 and M600pro. These platforms are able to fly for about 15 min with a payload of about 6 kg. Initially, System 13 was equipped with a barometric altimeter. However, the accuracy provided by such altimeter was not enough to keep the drone at a constant distance from surface; therefore, System 13 was upgraded with a more precise laser altimeter. All sensor devices are managed and controlled by an onboard computer where all recorded data are stored. Thanks to its multi-platform and multi-radar features, System 13 is used in various applicative contexts, such as for identifying archeological site foundations, search and rescue missions, landmine detection, etc.

Finally, we mention the recent UAV GPR prototype (System 14) proposed by the Department of Environment, Land, and Infrastructure Engineering of Polytechnic of Torino in 2021 [23]. System 14 is assembled by mounting the pulsed K2 IDS radar module on board a UAV platform based on the Venture VFF_H01 model, with a size of about 80 cm (height) × 2 m (width). The radar module is equipped with a 900 MHz antenna and a two-channel GPR acquisition unit. The UAV platform is able to fly for about 15 min, carrying a maximum payload of 7 kg. The navigation control unit is composed of a barometric sensor, a laser rangefinder, gyroscopes, 3-axes accelerometers, and a GNSS receiver. The Pixhawk 2 autopilot represents the central processing unit of the drone and elaborates navigation data to control the drone trajectory during the flight. The sensor data collection is managed by an onboard mini PC; this PC is also connected to a second base station computer through a Wi-Fi connection. On this base station PC, the mission planner software is run to remotely control the flight parameters (e.g., speed, altitude, remaining battery, etc.) in real time. System 14 is tested in the Alps region to assess its imaging capabilities for glacier monitoring and snow cover mapping.

A summary of the known UAV-GPR solutions is reported in Table 1.

**Table 1.** UAV GPR systems.

| System | Radar Technology | Frequency Range | Antenna | Measurement Configurations | UAV Platform |
|---|---|---|---|---|---|
| System 1 [3] | FMCW | Ka band | Linear array | MIMO | Small airplane |
| System 2 [4] | Pulsed Pulson P410 | 3.1–5.3 GHz | Helix | Bistatic (quasi-monostatic) | DJI Phantom 2 |
| System 3 [5] | Software-defined radio | Carrier frequency: 2 GHz (bandwidth not specified) | Antipodal Vivaldi antennas | Bistatic configuration with a 45° deg inclination | Hexacopter |
| System 4 [6] | Stepped frequency | 1–4 GHz | Horn | Bistatic (quasi-monostatic) | DJI Matrice 600 Pro |
| System 5 [8] | M-Sequence UWB radar sensor | System bandwidth: 5.05 GHz (0.95–6 GHz) | 1 TX custom designed spiral + 2 RX Vivaldi antennas | Two Vivaldi | 'Kraken' octocopter |
| System 6 [9] | Pulsed Pulson P410 | Frequency band: 3.1–5.1 GHz | Helix antennas | Quasi-monostatic | DJI Spreading Wings S1000+ |
| System 7 [10] | M-sequence UWB radar | 100 MHz–6 GHz | Two UWB Vivaldi antennas or two log-periodic antennas | Quasi-monostatic | DJI Spreading Wings S1000+ |
| System 8 [11] | Pulsed Pulson P440 | Frequency bandwidth: 3.1–4.8 GHz Carrier frequency: 3.95 GHz | Two log-periodic PCB antennas (Ramsey LPY26) | Quasi-monostatic configuration: down-looking | Self-assembled DJI F550 hexacopter |
| System 9 [15] | FMCW | 0.5–3 GHz | Vivaldi patch antennas | Bistatic (quasi-monostatic) | DJI Matrice 600 Pro |
| System 10 [16] | SFCW Planar R60 VNA | Selected frequency step: 10 MHz Selected frequency bandwidth: 500–700 MHz | Hybrid horn-dipole antenna transmitting and receiving, combining a tapered TEM horn and a half-wave dipole | Monostatic stepped-frequency continuous wave (SFCW) | X8 model made of 8 motors and 4 arms (2 motors per arm) from RCTakeOff |
| System 11 [17] | SFCW | 0.55–2.7 GHz | Hybrid Vivaldi-Horn antenna | Bistatic (quasi-monostatic) | DJI Matrice 600 Pro |
| System 12 [18] | Pulsed Cobra Plug-In GPR | 0.5–260 MHz | COBRA Plug-in SE-150 | Monostatic | DJI Matrice 600 Pro |
| System 13 [19] | Pulsed Cobra Plug-In GPR Cobra CBD Zond-12e | 0.5–1000 MHz | COBRA Plug-in SE-70 COBRA Plug-in SE-150 Cobra CBD 200/400/800 | Monostatic | DJI Matrice 600 DJI Matrice 600 Pro |
| System 14 [23] | Pulsed K2 IDS radar | Carrier frequency: 900 MHz (bandwidth not specified) | Not specified | Monostatic | Venture VFF_H01 |

## 3. Practical Issues and Technical Challenges for High-Resolution Imaging

UAV-based GPR systems offer many advantages such as high-rate surveys, minimization of hazards for operators, efficiency, and cost reduction; however, at the same time, they also suffer from some critical issues. In most cases, UAV platforms are affected by flight instability. This effect induces defocusing and localization errors in the radar images

that impair the detection performances [31]. Moreover, UAV-based GPR technology suffers from a reduced signal penetration with respect to its ground-based counterpart. Indeed, the penetration capability is limited by several factors, including additional propagation losses due to distance attenuation, diffuse scattering from rough terrain, as well as lateral and partial reflections. These issues are discussed more in detail in the following subsections together with some strategies for mitigating them.

### 3.1. UAV Constraint

Mini and micro UAVs are very compact platforms and so they can support only a limited payload mass. This entails limitations on the number and size of the electronic devices (e.g., antenna type) that can be embarked, as well as the capacity and the number of batteries, and so on. The payload constraint unavoidably poses various practical problems, such as a decreased imaging performance, a limited flight autonomy that reduces the size of the inspected area (i.e., the data throughput) and the radar transmission energy, and more difficult exploitation of sophisticated and accurate positioning system capable to estimate the UAV flight trajectory. The UAV payload optimization is an open technological challenge that still needs further investigation by the scientific community.

### 3.2. UAV Flight Dynamics

A critical point in UAV-based GPR surveys concerns the platform dynamics, which typically differs from the desired one due to the limited accuracy of onboard navigation sensors and the high sensitivity of mini and micro UAV platforms to wind gusts. Figure 6 shows a UAV trajectory performed by System 6 during a multi-line acquisition survey. As can be clearly seen, the UAV travels at a variable speed along non-straight measurement trajectories, thus leading to a non-uniform spacing of the measurement points.

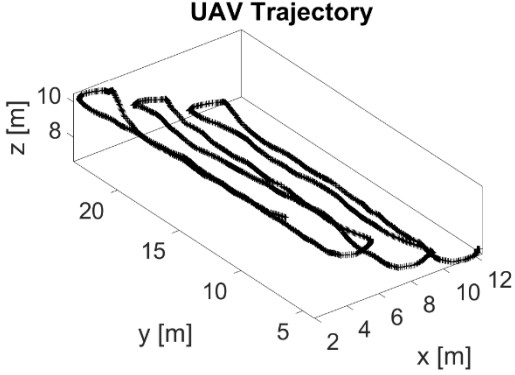

**Figure 6.** Example of an actual trajectory of a UAV-based GPR system during a multi-line survey.

When the measurement points are not equally spaced and the flight trajectory deviates from the ideal rectilinear one, standard GPR-focusing algorithms (e.g., migration [29,30]) cannot be applied, and more sophisticated imaging approaches accounting for UAV positioning information should be exploited [31].

A possible imaging approach is based on a motion compensation (MoCo) procedure. MoCo is an advanced signal-processing procedure that accounts for the UAV deviations with respect to the ideal flight track. This procedure was first introduced in the framework of airborne SAR focusing [33], and the concept was later exploited for UAV-borne radar imaging in [8,13]. MoCo procedure requires accurate knowledge of the UAV platform position and velocity. However, the quality of such an information depends on the accuracy of both embarked navigation sensors/systems and the deployed ground-based tracking devices. A typical UAV navigation system involves the use of IMUs with gyroscopes and accelerometers, magnetometers, and GNSS receivers. Note that the UAV GPR systems described in this review are equipped at least with one GNSS receiver (standalone solution). When the standalone GNSS receiver performance is not enough to guarantee a positioning

accuracy (at least smaller than half the radar signal wavelength), various solutions may be implemented. A first possibility consists in mounting onboard the UAV LIDAR altimeter, providing height information at a centimeter scale. LIDAR is typically integrated with navigation control units, which allows an estimation of the distance between the radar and the terrain to accurately follow the terrain topography (terrain-follow technology [3,5,9,13,19]).

When sophisticated altimeters are not available, alternative solutions have been devised. In particular, edge detection algorithms have been exploited in [4,11] to estimate the UAV altitude variation directly from the radar data. Edge detection is a post-processing procedure that allows compensating the height deviations with accuracy proportional to the wavelength of the radar signal. However, hardware solutions (such as LIDAR/range finders) and software strategies (such as edge detection) allow researchers to only compensate the altitude variation of the UAV without providing an accurate compensation of the horizontal deviations.

Accurate vertical and horizontal positioning information can be achieved through the exploitation of the CDGPS technique [34]. CDGPS implements the differences between the measurements collected by two relatively close GNSS receivers to filter out the common errors affecting them (i.e., satellite clock errors, tropospheric and ionospheric errors). In this regard, a typical configuration foresees at least one GNSS receiver placed onboard the UAV, and a second ground-based GNSS receiver. CDGPS data can be performed both in real-time (real-time kinematic—RTK) or in post-processing (post-processing kinematic—PPK). The RTK solution has been implemented in [6,8,9,15,17] whereas the PPK solution has been applied in [12–14]. Although the PPK strategy does not allow real-time control of the airborne trajectory, it is simpler and less expensive with respect to RTK, since it does not require any communication link between the GNSS receivers, still achieving very accurate offline trajectory estimation. Moreover, it is worth mentioning the sophisticated algorithm developed for System 3 to accurately control the platform trajectory during the data acquisition stage. Indeed, System 3 uses a very accurate UAV positioning control approach based on the backstepping method [35] and the desired angular acceleration function (DAF). The combined use of backstepping and DAF estimation methods allows exploitation of multiple sensor data (e.g., from IMU, GPS, and altimeter) to estimate the UAV flight dynamics in real time with high accuracy. As a result, the trajectory deviations can be limited to a few centimeters, thus allowing the collection of high-quality radar data.

### 3.3. Clutter

Clutter is another relevant issue in UAV-based GPR imaging applications. The term clutter is associated with all signal reflections occurring in the measurement time interval and interferes with the subsurface targets echoes. The main clutter source for GPR systems is the direct coupling between the transmitting (Tx) and receiving (Rx) antennas. This clutter is more significant when the antennas are very close to each other and a perfect isolation between them is not guaranteed. Indeed, part of the transmitted signal couples directly into the receiver without interacting with the targets.

The effect of direct coupling can be clearly distinguished from the air–ground interface when the UAV's flight height above the ground is large in terms of the probing radar wavelength (e.g., see raw data collected by System 8 in Figure 7). In these cases, direct coupling does not mask the signal backscattered from shallowly buried targets and it can be easily removed from the raw data by filtering procedures, such as background removal and/or time gating [2,36,37]. As is well known, background removal replaces the current trace (A-scan) with the difference between it and the average value of all A-scans (or a subset of A-scans close to the trace of interest) in the radargram (B-scan). However, background removal also acts on the signal scattered by the target; thus, further ground compensation procedures are required to restore the point spread function (PSF) of the system [38]. On the other hand, time gating selects the time interval containing the signals scattered from targets of interest and sets the signal outside such interval to zero. Figure 8 shows the effect of applying the time-gating procedure to raw data displayed in Figure 7.

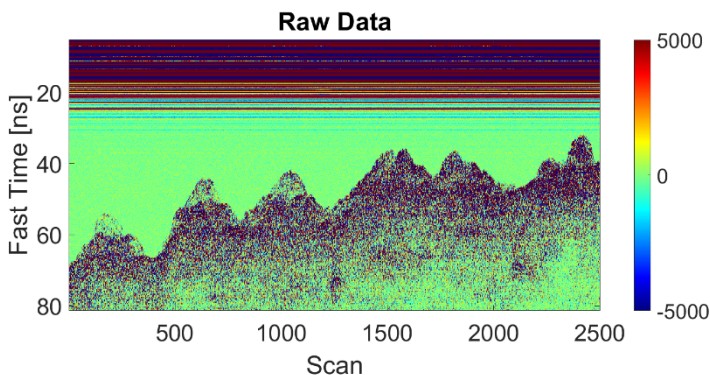

**Figure 7.** Raw data example collected by System 8.

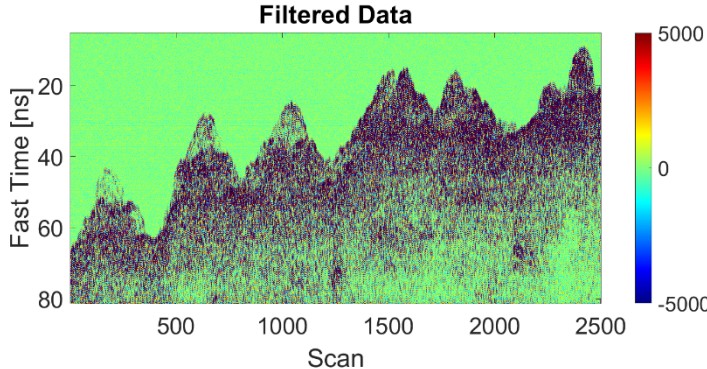

**Figure 8.** Filtered data after time-gating procedure.

The scattering from the air–soil interface is a more challenging source of clutter for UAV-borne GPR applications. Indeed, the signal reflection coming from the air–soil interface might completely obscure the backscattering signal associated with shallowly buried targets, thus limiting the detection performance. Time gating is the simplest strategy used to mitigate the reflection from the air–soil interface. However, the selection of the optimal time window containing the signals from useful targets represents a critical issue. Indeed, when the targets are close to the air–soil interface, the time gating may also erase part of the useful signals. Background removal is another popular filtering approach used to remove the effect of the air–soil interface. Note that this procedure works under the assumption that the clutter is spatially constant along the measurement direction, i.e., the air–soil interface is flat. The gating and background removal strategies are the most popular signal processing strategies used for mitigating the surface clutter and have been exploited by System 6 [9], System 7 [10], System 8 [11] and System 11 [17].

Another method for filtering surface clutter is based on the subspace projection [39,40]. This procedure assumes that the clutter energy is stronger than the energy backscattered by the target and, accordingly, its contribution is associated with a larger singular value of the raw data matrix. Therefore, once the singular value decomposition (SVD) of the data matrix is computed, the gathered data are projected on the subspace defined by the singular vectors and associated with the singular values different from the largest ones. A key point of this method is the estimation of the clutter subspace dimension, which is often based on the visual inspection of the curve of the singular values. The subspace SVD de-noising procedure has recently been implemented by System 7 [25] and System 8 [11].

Two further surface clutter mitigation strategies work with multi-antenna radar configurations. The first one was implemented in System 1 [3] by combining hardware and software solutions. Indeed, System 1 is characterized by a phased array antenna, which is able to narrow the antenna beam in the across-track direction, thus limiting the lateral surface returns. Then, in order to mitigate the residual clutter, a software solution based on a Doppler filtering procedure is also exploited. Indeed, all signal contributions coming from

off-nadir directions and collected over multiple radar scans produce a Doppler shift in the signals. Upon exploiting a zero-Doppler filtering procedure, System 1 is able to filter out the Doppler signals associated with off-nadir echoes, thus reducing the clutter disturbances. A second clutter mitigation procedure based on the multi-antenna concept was adopted by System 9 [15]. This last is characterized by multi-static antennas mounted on a flexible boom. The antennas were distributed along *z* and *x* directions to allow three-dimensional imaging and a large variety of bi-static combinations, thus enabling clutter mitigation capabilities. Indeed, by considering only one single radar image, due to the left–right ambiguity, it is not possible to estimate the depth of a target (see also [14]); however, by combining multiple radar images, i.e., those acquired at different look angles, the clutter contribution can be mitigated.

## 4. Applications

This section provides an overview of the main applicative contexts wherein formerly described UAV GPR prototypes are used. These include landmine detection, glaciers and snowpack investigations, soil moisture mapping, etc. We will address the advantages and the practical issues connected to the use of UAV-based GPR technology in these contexts and discuss the main experimental results reported in the literature.

### 4.1. Surface Object Detection

This subsection is concerned with applicative examples related to the detection of small objects placed on the surface of the surveyed scenario.

UAV-based GPR imaging results depend on multiple factors: flight parameters, measurement configuration, the quality of the data, as well as the adopted data-processing strategy. In order to assess the imaging system capabilities, several prototypes were firstly tested in a free-space scenario before proving their effectiveness in subsurface prospections.

As previously discussed, System 1 is a 3D imaging radar with no penetration capabilities. However, it was the first radar for a UAV operating in the sounder mode. The imaging capabilities of System 1 were firstly assessed by performing numerical tests. Indeed, in [41] simulations were performed to analyze the impact of the antenna vibrations on the 3D imaging quality. Then, a proof-of-concept laboratory test was carried out in [42] by performing the imaging of three trihedral corner reflectors. The first experimental test in labor-like scenario proved the imaging system capabilities. Unfortunately, imaging results associated with real datasets have not been reported.

The imaging capabilities of System 2 were assessed by performing tests in the presence of surface targets. Specifically, three experimental scenarios were reproduced by flying over three different targets [4]: a row of six trees, four vehicles in a parking lot, and two humans standing in a parking lot. During the first test, the system surveyed a row of six trees with different heights. The canopy of the tree was clearly visible in the collected radargram and the signal penetration throughout the canopy was also observed. As for the second test, backscattering signals by four different parked vehicles were examined. Differently from the tree canopy, the soil scattering signals were masked by the reflections from the vehicles. The last test regarded two people standing in different poses. In this case, the ground returns were more visible compared to the previous test and the radar returns from the human subjects were clearly observable in the collected range profiles. All three tests demonstrated the capabilities of the radar prototype to detect the considered targets.

System 4 [6] was also tested with surface targets by performing various experiments in down-looking, forward-looking, and circular configurations [43]. The first experiment was carried out by surveying an area containing two corner reflectors and several filled plastic cans. These simple experiments demonstrated the imaging capability of System 4. Then, System 4 was tested by flying over a meadow and a stone masonry, and by performing the imaging of two targets into a vertical domain (x-z plane). Recently, System 4 was upgraded with two additional antennas and some interesting experiments in interferometric configuration were conducted. The interferometric experiments aim to demonstrate the

capabilities of the system to filter out the ambiguities induced by the topographic variations of the terrain and to accurately detect small metallic objects buried into the soil [7].

Experimental trials regarding the imaging of surface targets were also performed with System 8 [11]. A proof-of-concept measurement campaign was carried out by performing two flights over five surface targets. Since the radar frequency band was 3.1–5.1 GHz, targets were placed above the air–soil interface. The tomographic image referred to the first survey allowed the detection of only four targets, while the second survey showed all five targets, allowing a satisfactory estimation of the distances among them and their quota above the ground. More recently, the same UAV GPR system was exploited in [14] to perform two surveys at different altitudes over two corner reflectors placed at a relative distance of 10 m between each other. One corner reflector was covered with a cardboard box. The imaging problem was formulated by considering a horizontal investigation domain and the focused images of the surveyed scenario showed that no ambiguities occurred in the images when the targets were illuminated at nadir. Conversely, false targets due to the left–right ambiguity appeared when the target was not illuminated at nadir. Further investigations were conducted with System 8 by performing the imaging in the vertical plane [13]. The latter study aimed at comparing the imaging performance when using UAV positioning information provided by standalone GPS and CDGPS, respectively. As expected, CDGPS position data allowed for better imaging compared to standalone GPS data, an estimation of the relative distance between targets, as well as target elevation above the ground with higher accuracy.

### 4.2. Landmine Detection

GPR is an important technology used for landmine detection since it is able to detect both metal and plastic landmines [44]. Moreover, it can also work in contactless mode by exploiting ground vehicles and UAV technology. Different UAV-based GPR prototypes were deployed for landmine detection. System 3 [5] was the first prototype used for landmine detection purposes. In order to assess the detection performance of the SDR-based GPR technology, many experiments were carried out with different types of landmines. Specifically, three landmine prototypes were buried: (i) a bottle-made artifact (with 20% of metal component), (ii) a fully metallic artifact, and (iii) a PBC tube-made artifact (with 30% of non-uniform metal component). The landmines were buried up to depths around 20 cm. In addition, two other types of metallic elements, acting as false landmines, were buried in the area. Fifteen surveys were performed in the presence of wet terrain (70% of humidity), which consequently made the full penetration of GPR signal difficult. The detection performances were assessed by calculating the Receiver-Operating Characteristic (ROC) curves. The results demonstrated that the UAV GPR system based on SDR technology could detect landmines made up of at least of 30% metal with a detection accuracy rate of 80%.

System 6 was tested for landmine detection applications. Early experimentations were conducted using a small and compact radar module, operating in the band 3.1–5.1 GHz [9]. Four experiments were carried out in different scenarios with different targets (metallic and dielectric). The main goal of the experimentation was to demonstrate the capability of the system to provide high-resolution underground images by exploiting high-accuracy trajectory information provided by the onboard UAV positioning systems. Recently, System 6 was upgraded with a different radar sensor operating in the frequency range of 100 MHz–6 GHz in order to enhance the penetration capability (System 7) [10]. Two flights in autonomous mode were also performed for testing innovative signal processing procedures. The tests showed improvements in terms of penetration as the system provided high-resolution 3D SAR images for a metallic disk with a 9 cm radius firstly placed in a small hole with an 8 cm depth without soil covering (first measurement), and then covered with soil (second measurement). Therefore, the achieved results demonstrated that the system can be very useful for landmine detection purposes.

System 9 was also developed for landmine detection but its first experimentations were carried out in non-operational scenarios [15]. Indeed, the system was initially set up in an Inverse SAR (ISAR) measurement configuration. The targets were placed on a turntable and the radar module was placed at different distances ranging from 2.5 m to 4.5 m. The scope of this measurement setup was to emulate circular UAV flights at different heights. In a first experiment, a Bakelite anti-tank mine with a 30 cm diameter and a fragmentation anti-personnel mine with a 11 cm length were placed on the turntable, while the ground-based radar system was at a fixed position, making the motion compensation much easier. The radar images demonstrated that the identification of a mine type should be possible even in the low frequency band of 0.5–3 GHz.

System 11 was specially designed for landmine detection purposes [17]. The first experiment was performed in labor-like scenario by mounting the sensor module on a motorized rail and burying an anti-personnel (AP) landmine of cylindrical shape, with a size of 8 cm $\times$ 14 cm into a polygon box with a depth of 20 cm. The scene was probed by moving the sensor with a constant velocity of about 0.6 m/s at a distance of 20 and 50 cm from the landmine. Then, a field measurement campaign was conducted in Skopje, North Macedonia. Two landmines were buried—the first one was the same AP landmine used in the laboratory test and was placed 20 cm deep into the soil, and the second one was a plastic AT landmine of cylindrical shape with a size of 27 cm $\times$ 13 cm which had its top aligned to the ground surface. The UAV survey was carried out by moving the sensor backwards and forwards with a velocity of about 0.6 m/s at a height variable between 10 to 50 cm above the ground. The imaging results showed that both landmines were visible into the radargram, thus demonstrating the detection capabilities of the system.

### 4.3. Soil Moisture Mapping

Monitoring soil moisture content is crucial for understanding hydrological processes, climate change, pollution assessment, etc. GPR was proven to be a useful technology for soil moisture measurements due to its high-resolution and non-destructive properties [2]. UAV-borne GPR is a cost-effective solution that can cover wide and not easily accessible regions, significantly reducing measurement efforts. In addition, thanks to its contactless working mode, it would not impact with plants and ground during the growing phases. For this reason, the UAV GPR system is becoming a very attractive technological solution for this applicative context.

System 10 was the first UAV-based GPR prototype, demonstrating the concept of soil moisture mapping [16]. Specifically, a full waveform inversion method was proposed to link the soil moisture content with the soil permittivity through the surface reflections. In order to show this, three surveys were performed in different agricultural fields, placed in the loess belt region of Belgium and characterized with different soil moisture distributions. These fields were chosen since their soil moisture content was mainly controlled by local topography. The radar operative bandwidth was set in the range of 500–700 MHz to avoid the effects of surface roughness. The flights were carried out by manually driving the UAV at an altitude between 1 m and 5 m. The achieved results showed that the soil moisture maps were in agreement with the topographical maps of the fields and aerial photogrammetry observations, thus demonstrating the potential use of a UAV-based GPR system used for precision agriculture and environmental monitoring purposes.

### 4.4. Snowpack Stratigraphy and Search Rescue

Stratigraphic information of the snowpack, such as depth, density, and layering, is crucial in snow resource management, which can affect public safety, hydropower production, and agriculture. Snowpack stratigraphy measurements are conducted by performing human surveys, and the density of measurements is typically obtained via samples collected through manually dug snow pits. These measurement methodologies are time-consuming and not always feasible, especially when large areas are mapped. GPR surveys are used in snowpack analysis to provide information on the density and depth

of the snow. However, GPRs are conventionally deployed on the ground by moving the radar antenna in direct contact with the ground. GPR can be mounted onto a snowmobile or manually moved for surveys over an undisturbed and flat snowpack. However, thanks to UAV platform flexibility, GPR prospections also become possible in more challenging scenarios (e.g., rough avalanche debris).

System 5 [8] was the first UAV-based GPR prototype developed for surveys of a layered snowpack over ground or sea ice. The system design was conceived with the aim of constructing a light and portable device with high resolution which can detect prominent snow layers. Two measurement campaigns were performed: the first test was carried out by surveying a transect in wet snow along a road; the second was a slow overflight over a buried person and a metal plate placed at different depths. The goal of these two trials was to assess the system's capabilities in resolving snowpack stratigraphy and detecting a person and a target.

In the first experiment, System 5 surveyed a road transect covered with snow by flying at an altitude of 50 cm. The recorded radargram clearly showed a high reflection about 40–50 cm below the radar antenna coming from the snow surface, while a significantly weaker reflection from the ground surface was visible at a depth of 160 cm. This last reflection was associated with a snowpack depth of 120 cm. Moreover, four clear reflections within the snowpack indicated the transition between different snow layers, thus mapping the snowpack stratigraphy as well as ice layers. In the second trial, System 5 was used with search and rescue purposes. Indeed, the test involved monitoring a snowy area, as well as hiding a buried person and metal plate at different depths. During the test, four surveys were performed over a human and a metal targets. The metal plate was present in all four passages, while the person was detected only in one flight. This last experiment demonstrated the penetration and imaging capabilities of System 5 and its potential for search and rescue missions.

System 14 was specifically designed for snow cover mapping [23]. Two experimental tests were performed during the winter of 2020–2021 in Valle d'Aosta, Italy. The first experiment was conducted at the remote basin of Cheneil—an area of great importance for the valley since it hosts the main sources of water, while the second experiment was performed at Gressoney in a fenced zone close to a ski-resort area. During the first test, two ground-based GPR surveys were performed for assessing the imaging capabilities of the UAV-based GPR system. In particular, firstly, the ground-based data were calibrated by measuring the snow depth and exploiting a graduated rod. This allowed estimation of the average wave velocity by comparing the travel times of the GPR signals and the punctual snow depth measurements. Then, in order to compare the ground-based data versus the air-based data, the range delay due to distance between the drone and air–snow interface was compensated from the UAV GPR radargram. Nevertheless, during the second test, the ground-based GPR data were not available; therefore, to assess the drone-base dataset, the radar signals coming from the air–snow interface were compared with the range finder data. The good agreement between the ground-based and air-based radargram demonstrates the snow mapping capabilities of the UAV-based GPR prototype.

## 5. GPR Data-Processing Approaches

GPR data processing is a broad research area, as confirmed by many papers and books, e.g., [2,36,37] and references therein. A comprehensive review of these techniques would be too long and out of the scope of the present paper; thus, a brief discussion about the main approaches relevant to UAV-borne GPR systems is reported below.

GPR data-processing techniques can be divided in two groups:

- standard processing;
- advanced imaging/focusing algorithms.

It should be pointed out that the techniques belonging to the second group are typically carried out after the application of the standard processing.

### 5.1. Standard Processing

Standard processing refers to classical filtering procedures aimed at mitigating clutter, removing noise interferences, compensating motion errors, and signal attenuation. Conversely, advanced imaging strategies aim to provide a clear focused image of the inspected area.

Most of the standard GPR procedures, such as time gating, background removal, gain, and so on, were implemented in the majority of UAV-GPR systems described in Section 2. In particular, System 1 adopts sophisticated filtering strategies for mitigating clutter thanks to antenna beamforming. System 2 implements a correction of altitude variations by aligning the radar traces based on the signals reflected from the air–soil interface. However, when surface objects (i.e., canopy) are present, returns from the air–soil interface cannot be estimated. To overcome this issue, System 2 performs an interpolation procedure to reconstruct the returns from the ground. System 3 applies standard filtering and gain procedures for mitigating the clutter, a sophisticated algorithm to control accurately the UAV trajectory during the survey, and a detection algorithm based on a matched filter to maximize the signal-to-noise ratio (SNR). The detection algorithm produces a detection map in order to identify the landmines. System 5 adopts a signal strategy similar to that used by System 3 since it correlates the radar data with the transmitted pulse to produce the impulse response of the medium within the radar range. Additional processing is also performed in order to compensate the nonlinear antenna effects and the noise disturbances. Finally, a histogram equalization and thresholding procedure is also considered to detect the main target features. Systems 6, 7, and 8 apply standard gating, background removal, and clutter mitigation procedures before executing advanced imaging algorithms. System 10 and 12 implement standard background removal to filter both antenna coupling and the signal reflection from the air–soil interface.

### 5.2. Advanced Imaging/Focusing Algorithms

After the standard processing, advanced focusing procedures are implemented to enhance the resolution and the interpretability of GPR images. As is well known, a focusing algorithm processes radargrams in order to transform point targets, appearing as diffraction hyperbolas, into bright spots [2,36].

In order to describe the UAV-borne GPR imaging problem, let us refer to the 3D scenario depicted in Figure 9. A monostatic radar mounted onboard the UAV illuminates the scene in the down-looking mode. The radar transceiver collects the signals scattered from the scene over the angular frequency range $\Omega = [\omega_{min}, \omega_{max}]$ while moving along the flight trajectory $\Gamma$. This is supposed to have an arbitrary shape in space and each measurement point is described by the position vector $r_m = x_m\hat{x} + y_m\hat{y} + z_m\hat{z}$. The targets are located inside the investigation volume $D$.

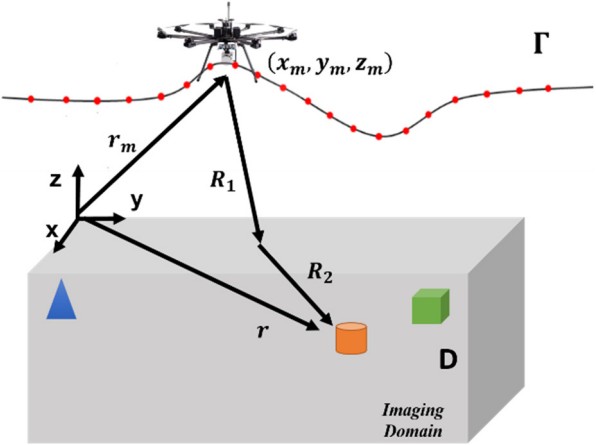

**Figure 9.** UAV-borne GPR imaging scenario.

The radar sensing process is here described by the linear scattering model holding under the Born approximation, i.e., the mutual interactions between the targets in $D$ are neglected [45]. Moreover, a scalar model is adopted for the sake of simplicity and then the measured scattered field at each point $r_m$ is expressed by the following linear integral equation [46]:

$$E_s(\mathbf{r_m}, \omega) = k_s^2 \int\int\int_D g_e(\mathbf{r_m}, \mathbf{r}, \omega) E_i(\mathbf{r}, \omega) \chi(\mathbf{r}) dr \tag{1}$$

where $E_s$ is the scattered field, $E_i$ is the incident field in $D$, $\chi(\mathbf{r})$ is the unknown contrast function at any point $\mathbf{r} = x\hat{x} + y\hat{y} + z\hat{z}$ in $D$, $g_e$ is Green's function of the scenario, and $k_S$ is the propagation constant in the soil. Note that different approaches were described to evaluate the incident field $E_i$ and the Green's function $g_e$ with reference to a two-dimensional half-space scenario [31].

Various algorithms have been proposed in recent years to solve the GPR imaging problem. The most popular procedures are Kirchhoff's wave–equation [47] and frequency–wave number $(\omega, k)$-based migration algorithms [29,30]. These processing methodologies were originally formulated for seismic imaging applications and then adapted to SAR imaging [48]. In spite of their computation effectiveness, a major limitation of migration algorithms is the key assumption that radar data are collected along a rectilinear measurement trajectory. This assumption is almost never verified during UAV-based GPR surveys. As a result, most focusing algorithms exploited in the UAV GPR context are essentially the SAR back-projection or beamforming algorithms (e.g., [9]). These last perform a coherent integration of the radar echoes recorded by the sensor along the along flight track to produce a reflectivity map of the scene, i.e.,

$$\chi(\mathbf{r}) = \int_\Gamma \int_\Omega E_s(\mathbf{r_m}, \omega)(g_e(\mathbf{r_m}, \mathbf{r}, \omega) E_i(\mathbf{r}, \omega))^* d\mathbf{r_m} d\omega \tag{2}$$

where the symbol * is the conjugation operation.

Unlike migration, the back-projection method can handle arbitrary flight trajectories (e.g., circular, spiral, etc.), even with non-uniform data spacing [14,26]. It is also worth pointing out that SAR-like algorithms are able to manage multiple-input multiple-output (MIMO) radar configurations [49]. However, their computation complexity is generally higher compared to migration since efficient implementations based on the fast Fourier transform (FFT) algorithm are not possible. SAR-like imaging procedures are exploited in the majority of the UAV-based GPR systems, as in System 1 [3] System 4 [6] System 6 [9], System 7 [10], System 8 [14], System 9 [15] and System 11 [17]).

Microwave tomography (MWT) is a special class of focusing algorithms based on the solution of an electromagnetic inverse scattering problem [36,37]. Differently from SAR-like algorithms, MWT procedures are inverse filtering methods that solve the imaging problem by inverting a mathematical model as the linear integral equation in Equation (1). Specifically, the imaging problem is faced as the solution of the linear inverse problem [50]:

$$E_s = L\chi \tag{3}$$

with $L : \mathcal{L}^2(D) \to \mathcal{L}^2(\Gamma \times \Omega)$ being a linear operator mapping the unknown space into data space, where both spaces are square-integrable function spaces. The inverse problem in Equation (3) is ill-posed and a regularization strategy is necessary to obtain a stable solution [50]. For example, the data inversion can be performed by resorting to the truncated singular value decomposition (TSVD) scheme [50]:

$$\widetilde{\chi} = \sum_{n=1}^{K_t} \frac{\langle E_s, u_n \rangle}{\sigma_n} v_n \tag{4}$$

where $\langle, \rangle$ is the inner product in data space; $\{\sigma_n, u_n, v_n\}_{n=1}^\infty$ is the singular spectrum of the operator $\boldsymbol{L}$; $\sigma_n$ are singular values; and $u_n$ and $v_n$ are orthonormal basis functions in the data space and unknown space, respectively. The regularization parameter $K_t$ is the number of

retained singular values, whose value is typically fixed in such a way to find a compromise between resolution and stability for the solution. The modulus of the regularized contrast function $\widetilde{\chi}$ in Equation (4) defines a spatial map referred to as tomographic image.

MWT generally provides enhanced performances than SAR-like approaches as it produces a more robust solution w.r.t the noise on data. Recently, MWT approach has received attention in UAV-based GPR imaging owing to its flexibility in dealing with arbitrary measurement configurations (e.g., multibistatic, multi-view/multi-static, etc.), data collection geometries, and background scenarios (e.g., rough terrain profiles). This imaging approach was exploited by System 8 in [11,13].

Finally, full-waveform inversion is another advanced signal-processing approach that is worth mentioning [16]. Full-waveform inversion accounts for a specific mathematical model, which links the collected radar data with the soil electromagnetic properties in the surveyed area. This procedure aims at retrieving the soil properties (e.g., permittivity) by minimizing a cost function accounting for the "distance", in the data space, between the observed data and model data. Full-waveform inversion was implemented in System 11 to provide an estimate of the soil permittivity and, subsequently, to generate a spatial map of the water content.

## 6. Experimental Tests

This section describes two experimental tests showing the potential of down-looking UAV GPR systems. The first example concerns the imaging of two small metal targets placed on the surface of the surveyed scenario (*surface imaging example*). This test aims to assess the imaging capabilities of System 8 by exploiting two focusing strategies based on the standard back-projection and MWT imaging approach, respectively. The second example regards the detection of a metal plate buried into the soil (*subsurface imaging example*) to assess the penetration capabilities of a commercial UAV-based GPR system [21].

### 6.1. Surface Imaging Example

A proof-of-concept measurement campaign was performed to assess the imaging performance of System 8 when the standard back-projection and the MWT imaging approaches were implemented to focus the radargrams. Two surface metallic objects were present during the trial and one of them was hidden under a cardboard box. Further details about the measurement campaign and the measurement configuration are reported in [14]. The implemented signal-processing pipelines are shown in Figure 10 (see also [12,14] for further details).

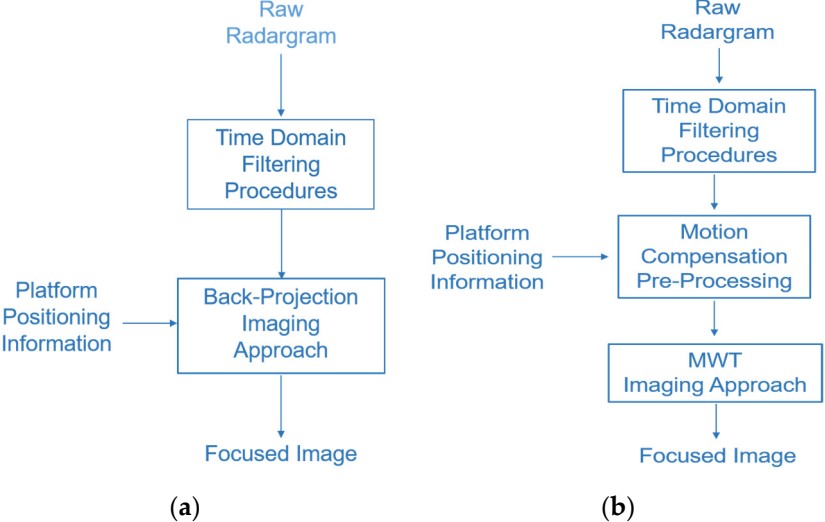

**Figure 10.** UAV-GPR signal-processing schemes: (**a**) back-projection imaging-based strategy; (**b**) MWT imaging strategy.

The first imaging strategy is represented by the block diagram in Figure 10a and herein referred to as Strategy A. As can be seen, the raw data are pre-processed in the time domain by using standard filtering procedures for clutter mitigation, such as time gating and background removal. Then, the filtered radargram is transformed in the frequency domain by a Fourier transform operation, and, finally, the back-projection imaging approach (see Equation (2)) is applied by accounting for the UAV position information provided by onboard navigation instruments.

The second imaging strategy (Strategy B) is sketched in Figure 10b. Strategy B differs from Strategy A since it implements a MoCo procedure before the focusing step. MoCo exploits the positioning information provided by the navigation sensors and compensates the distortions in the radargram by performing a range alignment and along-track correction procedures. The range alignment compensates the UAV altitude variations which occurred during the flight with respect to a constant reference flight altitude; the along-track correction interpolates and resamples the data in order to evenly obtain spaced data along the track. After the MoCo step, data are transformed into the frequency domain and processed according to the MWT approach formerly described in Section 5.2 instead of applying the back-projection technique. We would like to remark that the MoCo procedure could be also applied with the SAR back-projection imaging approach; on the other hand, the compensation of motion errors within the MWT approach could be achieved by accounting for UAV positioning data directly during the inversion stage.

The investigation domain is the plane defined by the $x$ axis and the $z$ axis, where the $x$ axis is the along-track direction and the $z$ axis is the depth.

The hardware components of System 8 are detailed in Section 2. However, during the trial, two GNSS receivers were exploited: one receiver was installed onboard the UAV and the second receiver was used as a ground-based station for implementing the CDGPS positioning estimation technique.

The field trial was carried out on 5 July 2019 at an amateur UAV flight test site located in Acerra, Naples, Italy. As shown in Figure 11, two metallic trihedral corner reflectors ($T_1$ and $T_2$), with size 0.4 m × 0.4 m × 0.75 m, were the targets of interest. Both corner reflectors were placed on the ground and $T_2$ was covered with a cardboard box. The targets were aligned along a straight line at a distance of 10 m between each other. The flight test had a duration of 21.54 s and covered a 17 m-long path; along this track, data were gathered at 190 measurement points that were not evenly spaced. The radar parameter settings adopted during the measurement campaign are summarized in Table 2, while Table 3 reports the main signal-processing parameters.

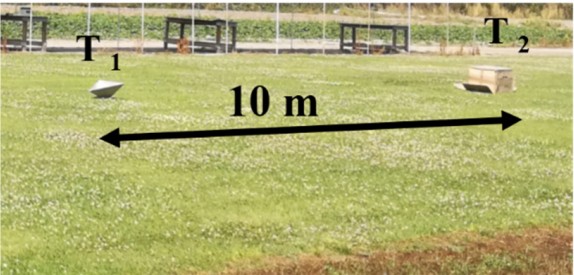

**Figure 11.** Experimental test site: surface imaging scenario.

**Table 2.** Radar system parameters.

| Parameters | Specification |
| --- | --- |
| Carrier frequency | 3950 MHz |
| Frequency band | 3100–4800 MHz |
| Pulse repetition frequency | 14.28 Hz |

**Table 3.** Signal-processing parameters.

| Parameters | Specification |
|---|---|
| Time-gating window | 24–40 ns |
| Frequency range | 3100–4800 MHz |
| Frequency step | 30 MHz |

Figure 12 shows the raw radargram (Figure 12a), the filtered data (Figure 12b), and the radargram obtained after MoCo and filtering procedures (Figure 12c), respectively. In Figure 12a, two diffraction hyperbolas corresponding to $T_1$ and $T_2$ are clearly visible and their apexes appear at slow times equal to 4 s and 9 s, respectively. The strong horizontal constant signals represent the Tx-Rx antenna coupling. Despite this large clutter contribution, the radar is capable of detecting both targets and can recognize that $T_2$ is partially hidden by a weakly scattering object. This claim is suggested by the weaker hyperbola appearing before the hyperbola produced by $T_2$ along the fast-time axis.

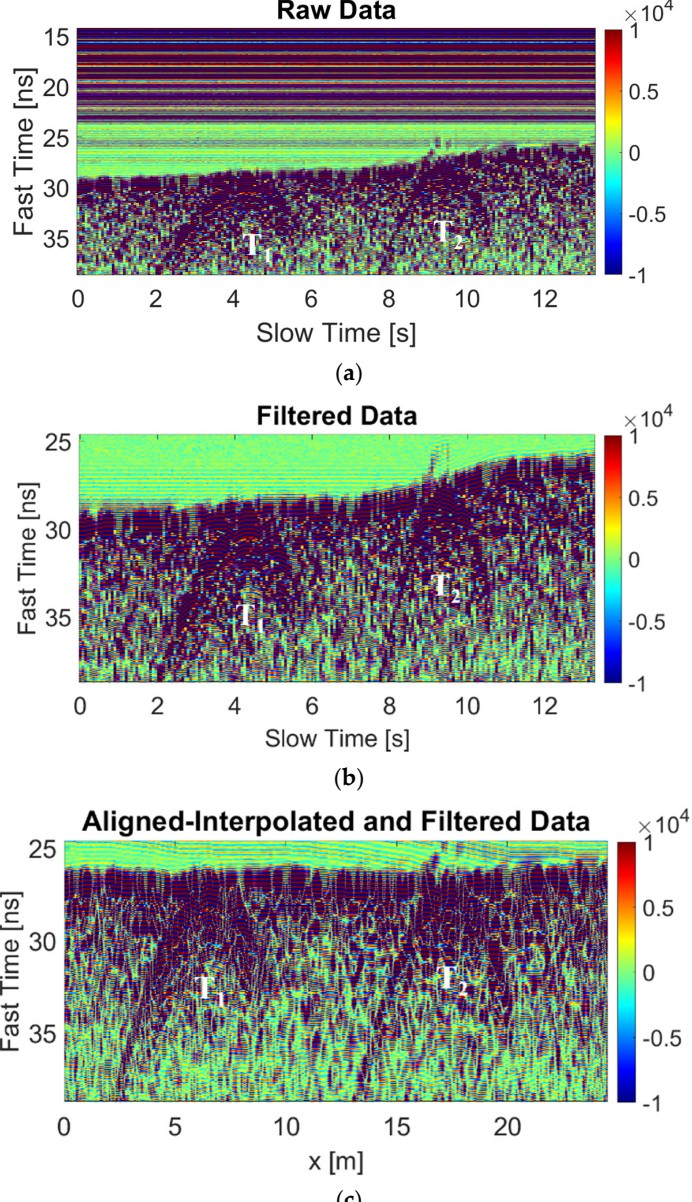

**Figure 12.** UAV-GPR data: (**a**) raw data; (**b**) filtered data; (**c**) radargram after MoCo and filtering versus traveled distance.

The filtered radargram shown in Figure 12b was achieved by applying the background removal and time gating. As clearly visible, the direct coupling signals are now strongly mitigated. We remark that the trajectory deviations alter the radargram shown in Figure 12b. Indeed, although the terrain profile of the test site is flat, the reflection from the ground exhibits a strong variation along the survey direction. This outcome is attributable to the altitude change in the UAV platform during the flight.

Figure 12c demonstrates the ability of the MoCo procedure to compensate the flight altitude deviations since the air–soil interface now appears almost flat. Moreover, thanks to the clutter mitigation filtering, the small apex above the second hyperbola (induced by the presence of the cardboard box) now becomes more evident than in Figure 12a.

The focused images achieved by applying Strategies A and B are depicted in Figure 13a and Figure 13b respectively. As can be seen, both imaging strategies reveal the presence of two bright spots related to targets $T_1$ and $T_2$, which can be easily discriminated with respect to the air–soil interface. Both images allow a correct localization of the targets by preserving their relative distance along the track. Indeed, the estimated distance between the two bright spots is about 10 m. The reconstruction depicted in Figure 13b achieved via the MWT imaging approach confirms the effectiveness of the MoCo step in correcting the UAV trajectory deviations and achieving a satisfactory target focusing. Furthermore, the result appears better focused with respect to that obtained using the back-projection strategy since the spots in Figure 13b are more concentrated and the air–soil interface is better defined and less noisy compared to Figure 13a. The imaging performance of MWT, which was better compared to the back-projection approach, has been demonstrated in previous works. Indeed, MWT attempts to solve the linear inverse scattering problem in a more rigorous way with respect to SAR back-projection [51].

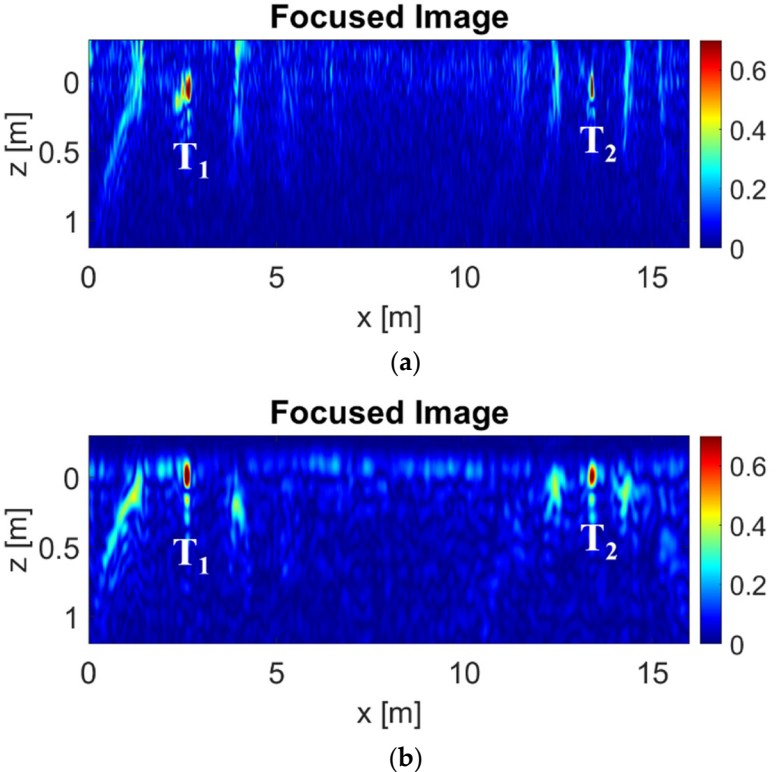

**Figure 13.** Focused images: (**a**) back-projection imaging; (**b**) microwave tomography-based imaging.

*6.2. Subsurface Imaging Example*

An experimental test assessing the penetration capabilities of UAV-based GPR technology was recently carried out by the authors. The test deals with the detection of a metal plate with a size of $0.25 \times 0.35$ m, which was buried into the ground at a depth of

0.3 m. The UAV-GPR system was assembled by Novatest s.r.l., Falconara Marittina, Italy (https://www.novatest.it/, accessed on 9 June 2022), based on the commercial Cobra CBD radar module [21] and the DJI Matrice 600 Pro platform. The radar module works at the center frequency of 500 MHz with a bandwidth variable in the range of 50–1400 MHz. The UAV autopilot was composed of an IMU and a GNSS receiver used to manage and control the UAV flight trajectory. The drone was also equipped with a laser range finder to enable the functionality of the terrain, and an additional GNSS receiver was connected over a wireless link to a ground base station by implementing the CDGPS technique. GPR and UAV telemetry data were integrated and synchronized via a sky-hub PC data logger.

The field trial was carried out on the 12 May 2022 over an open area in close proximity of the Hydrogeosite laboratory of IMAA-CNR, located in Marsico Nuovo, a small town in Basilicata, Italy. Figure 14a shows the aerial view of the site provided by the Zoom Earth web application [52] with the UAV flight trajectory (red line) and the superimposed target (black rectangle). Figure 14b shows a photo of the metal plate taken during the burial operation. The main radar system parameters adopted for the data acquisition are described in Table 4.

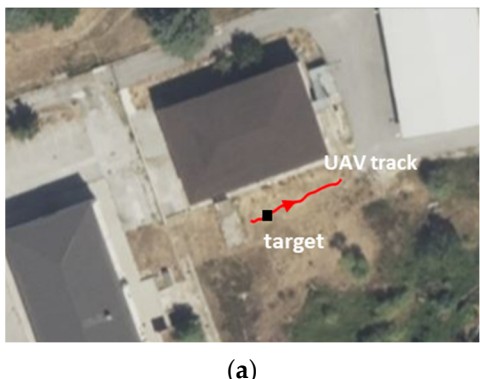
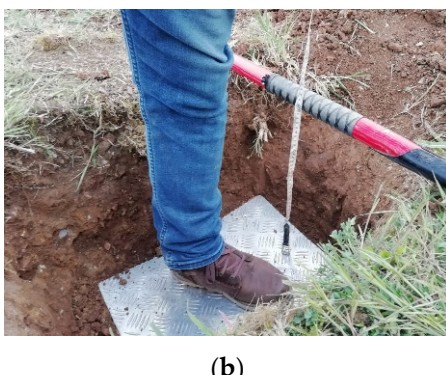

(**a**)                                                                                          (**b**)

**Figure 14.** Experimental test site: (**a**) aerial view of the area with UAV flight trajectory (red line) and target (black rectangle) superimposed; (**b**) photo of the buried metal plate.

**Table 4.** Radar system parameters.

| Parameters | Specification |
| --- | --- |
| Carrier frequency | 500 MHz |
| Frequency band | 600 MHz |
| Pulse repetition frequency | 33 Hz |

The UAV was manually piloted and a single passage was carried out over the area at a nearly constant altitude of 0.5 m. The flight lasted around 21.5 s and 450 radar traces were recorded over a 15.9 m-long track at an average speed nearly equal to 0.7 m/s.

The radar data were processed according to the signal processing scheme introduced in Figure 10a. Specifically, the raw radargram was firstly pre-processed in the time domain by setting the zero time and performing the time-gating operation, and then the filtered radargram was transformed in the frequency domain using a Fourier transform operation. Finally, the focusing procedure based on the back-projection imaging approach was applied by accounting for the signal propagation through the air–soil interface. In this regard, the Green's function $g_e$ of the half-space scenario involved in Equation (2) was evaluated according to the equivalent permittivity model formulated in [53].

The raw radargram collected by the system is depicted in Figure 15 where the horizontal axis is the trace index, while the vertical axis is the fast time. The raw radargram shows strong horizontal constant signals linked to the Tx-Rx antenna coupling. Despite this large clutter contribution, the radar also reveals the presence of some subsurface layers in the fast time window of 10–25 ns. Moreover, a weakly scattering anomaly associated with the buried metal plate appears around the trace 140 at the fast time of 14 ns.

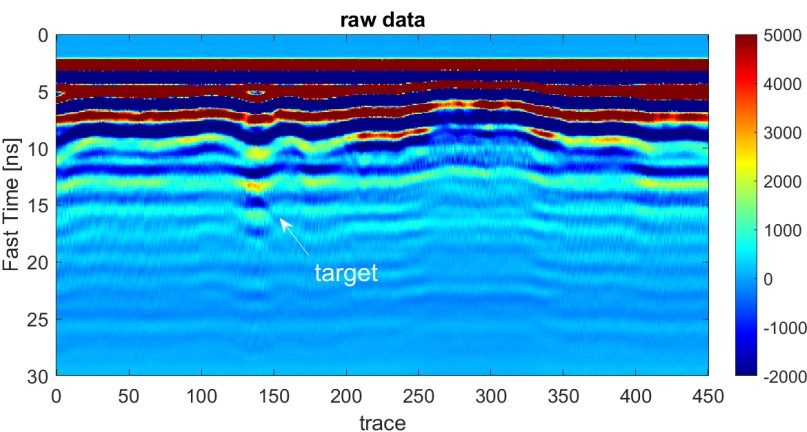

**Figure 15.** UAV GPR raw radargram.

Figure 16 shows the filtered radargram achieved after the application of the time-gating operation, which sets the signals outside the time interval of 8–30 ns to zero. Now, the air–soil interface and the target anomaly are better emphasized compared to their counterparts in the raw radargram (see Figure 15).

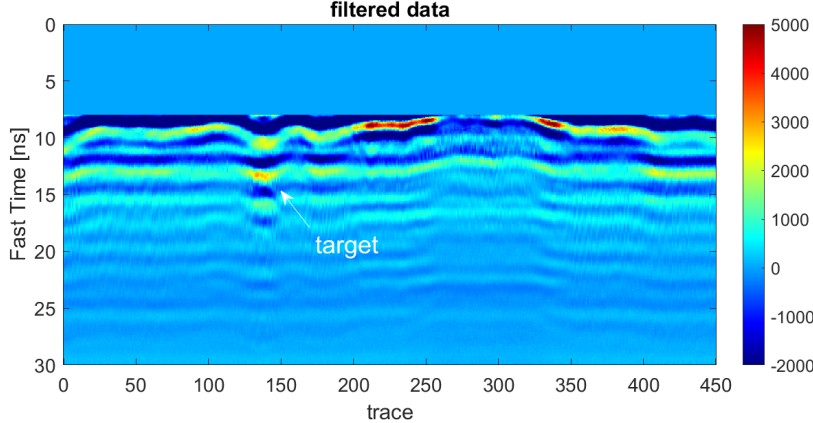

**Figure 16.** UAV-GPR filtered radargram.

All the parameters adopted for the radar data processing are listed in Table 5. Specifically, after setting the zero time at 2 ns, the filtered data in Figure 16 were transformed into the frequency domain and then focused via back-projection by assuming a soil relative permittivity of $\varepsilon_r = 16$. This value is justified by the fact that the soil was wet (e.g., see material properties in [2]) due to the rain that fell the day before the test. In order to reduce the computation effort related to the focusing procedure, the shift and zoom processing strategy described in [13] was applied by considering a synthetic aperture of 2 m.

**Table 5.** Signal-processing parameters.

| Parameters | Specification |
|---|---|
| Zero time | 2 ns |
| Time-gating window | 8–30 ns |
| Background relative permittivity | 16 |
| Frequency range | 200–800 MHz |
| Frequency step | 18.75 MHz |

Figure 17 shows the focused image of the surveyed area. This image was obtained by considering a rectangular investigation domain $D$ with a size of 13.6 m × 1 m, whose $x$ axis and $z$ axis correspond to the along-track direction and depth, respectively. The domain

$D$ was evenly discretized in square pixels with size of 0.02 m × 0.02 m. As can be seen, the focused image can clearly identify the shallow soil stratigraphy as well as the target that appears to be located at a depth around 0.3 m. Despite the simple proof-of-concept example, the achieved result demonstrates the potential of the considered UAV-GPR system to detect buried objects.

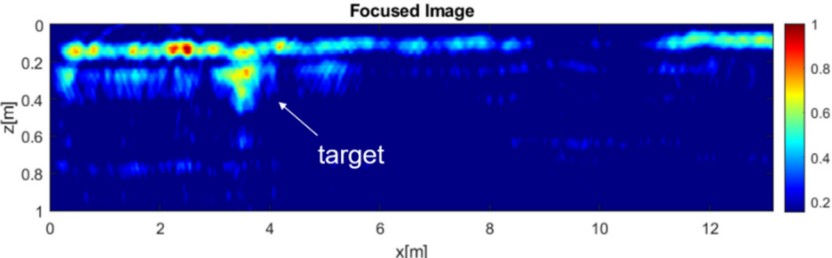

**Figure 17.** Focused image.

## 7. Discussion and Future Perspectives

This manuscript provided an overview of technological and practical aspects related to down-looking UAV-based GPR systems. State-of-the-art solutions, in terms of radar technology, UAV platform, and instrumentation, were described. Then, the main practical issues concerning UAV-based GPR technology, such as the payload constraints, the effects of UAV flight dynamics, and the signal propagation issues, were considered. The main strategies used to cope with the UAV flight dynamic issues are based on sophisticated devices, e.g., laser altimeters and/or multiple GNSS receivers, which help to estimate accurate navigation data. As for signal propagation issues, the main disturbances are associated with antenna coupling, surface clutter reflections, and electromagnetic interferences. To mitigate these effects, current UAV GPR systems take advantage of different filtering procedures.

An overview of the main applications was also reported with the aim of providing a clear picture of the effective potential of UAV-GPR technology.

A short description of the principal signal-processing strategies, aimed at providing a clearer and more interpretable radar image, was also reported. These strategies were grouped in two classes: SAR back-projection methods and MWT imaging approaches.

Two proof-of-concept experiments were also presented: (i) a surface imaging example and (ii) a subsurface imaging example. In the first experiment, two surface targets were detected by a UAV GPR prototype assembled by the authors. SAR back-projection and MWT imaging were implemented and both provided a satisfactory target localization, with MWT showing slightly better resolution performance than back projection. The second example proved the detection capabilities of a commercial GPR system in the case of a metal plate buried 30 cm below the ground.

In spite of the technological advances and the availability of sophisticated radar signal-processing algorithms, some scientific and technological challenges still need to be faced for the successful application of UAV GPR systems in operative conditions. The first issue regards the development of innovative solutions which can improve radar imaging performance in terms of sensitivity, clutter mitigation, and multipath. In this respect, some advances such as differential GPR [54] and subspace filtering methods [39,40] were made for ground-based GPR systems.

Differential GPR configuration can filter the surface reflection directly during the survey by recording the difference of the signals received by two antennas located at the same height above the air–soil interface and arranged symmetrically with respect to a central transmitting antenna. On the other hand, the subspace filtering method is based on principal component analysis. This technique performs the SVD of the raw data matrix and the clutter contributions are filtered out by assuming that the clutter energy is stronger than the energy backscattered by the target, and, accordingly data associated with the larger eigenvalues are discarded. Regarding multipath induced by interactions among the targets, ad hoc mitigation strategies should be developed to eliminate false targets

and improve the interpretability of the images. Therefore, following the example of the advances reported in [55,56], similar methodologies may be also developed for UAV-based radar imaging technology.

Another practical issue for UAV-borne GPR imaging is the need for processing large-scale data (in terms of probing wavelengths). To face this issue, an efficient data-processing method based on the shifting zoom approach has been proposed in [57] and later applied to the processing of large-scale airborne GPR data [58]. However, efforts are still required to update the methodology in order to address full 3D subsurface imaging.

Finally, a further challenge regards the design of innovative light and compact antenna arrays to implement measurement configurations different from the classical monostatic one in real time. Indeed, the use of antenna arrays, enabling multi-view and multi-static observations, is expected to increase the quantity and quality of data and therefore enhance the imaging performance.

**Author Contributions:** Conceptualization, C.N., G.G. and F.S.; methodology, C.N., G.G. and F.S.; formal analysis, C.N., G.G. and F.S.; investigation, C.N., G.G., G.E., G.L., G.F. and L.C.; writing—original draft preparation, C.N., G.G. and G.L.; supervision, C.N., I.C., G.G. and F.S.; project administration, I.C. and F.S.; funding acquisition, I.C. and F.S. All authors have read and agreed to the published version of the manuscript.

**Funding:** This research received no external funding.

**Acknowledgments:** The authors would like to thank the University of Oviedo, Gijón, Spain; the Department of Industrial Engineering (DII) at the University of Naples "Federico II"; Deutsches Zentrum für Luft- und Raumfahrt (DLR), Germany; the University of Maribor, Maribor, Slovenia; Dyrecta Lab, Conversano, Italy for providing us the pictures of their UAV GPR systems, showed in this manuscript. The authors would like to thank Gregory De Martino of CNR-IMAA for his support during the experimental tests.

**Conflicts of Interest:** The authors declare that there is no conflict of interest.

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
