# Peer review of "An Overview on Down-Looking UAV-Based GPR Systems"

_remotesensing, doi:10.3390/rs14143245_

Round 1
Reviewer 1 Report
The contribution claimed by the authors of this manuscript:
“This article provides a comprehensive review on the current state of the art and challenges related to UAV-based Ground-Penetrating Radar (GPR) imaging systems. First, a description of the available prototypes is provided in terms of radar technology, UAV platform and navigation control devices. After, the paper addresses the main issues affecting the performance of UAV-based GPR imaging systems such as the control of UAV platform during the flight to collect high quality data, the necessity to provide accurate platform position information in terms of probing wavelength, and the mitigation of clutter and other electromagnetic disturbances. A description of the major applicative areas for UAV GPR systems is reported. Furthermore, the main signal processing approaches currently adopted are detailed and two experimental tests are also reported to prove the actual imaging capabilities. Finally, open challenges and future perspectives are discussed.”
The English writing, organization and presentation quality of this work is good. Abstract and Discussion provide a very clear idea about this work and authors’ contributions. It is a really interesting research survey, containing very updated reference literature. I have noticed some issues which authors must address to improve the quality of this work.
1. In Introduction, it is highly recommended to provide a Table comparing your work with existing reviews or surveys. From Tables, readers can fastly know the contributions of each study.
2. Please remove some keywords.
3. Line 49-50: Please check the reference sequence and make relevant corrections against any inconsistency.
4. Line 93: Make the relevant corrections in writing format to adjust full visibility of text. I have noticed the same issue at several places line such as line 126, 160, 334 etc.
5. Section 1 is nicely presented. I have enjoyed writing fluency and easy to follow this work.
6. Try if you can provide Figure 9-10 in high resolution with more better quality.
7. In start of each section, providing a precise description is really nice which some review articles usually ignore.
8. Line 108: Authors have suddenly used new terms mini-UAVs and micro-UAVs. You can mention any specific parameters e.g., size, weight or range within () so readers can easily understand the difference.
9. Figure 5 does not look clear. It is recommended to provide figures with high resolution for a nice view and better readability of article.
10. Table 1 is nicely presented. Readers can easily understand the differences between different UAV-GPR solutions.
11. Section 3: It is suggested to add few more technical challenges along with relevant practical considerations.
12. UAV-based GPR system has any applicability to powerline or transmission line inspection?
13. Please recheck each equation and fully define each variable used in these equations as every reader is not of the same understanding level.
14. Figure 8: You can consider a different color rather than white for inside text.
15. I suggest making Discussion and future perspectives section more precise and short. Avoid repeating same lines, just focus on your key contributions, how this study can be helpful for relevant research fraternity and some possible future research directions.
16. Adjust writing format of COI statement.
17. For reference: I’m not sure if you can put direct URLs without providing more information such as Accessed date. Please refer to journal template.
Reviewer 2 Report
The paper is a review of the down-looking UAV-based GPR systems, proposing a description of most important milestones in terms of prototypes, a brief description of application fields with references to published articles, the analysis of the most important issues induced on GPR data by UAV acquisition, a brief description of GPR processing and two case studies to test the previously described topics. The language is generally fine and the exposed themes are of interest for the readers of Remote Sensing.
I have some minor remarks for the authors:
On page 4, lines 175-176: "recently" is repeated twice;
On page 7, line 296: "attitude" -> altitude;
On page 11, line 367, "attitude" -> altitude;
On page 14, lines 550-552: please verify the correctness of the double references to System 6 and System 7;
On page 19, line 749: "w.r.t.", please write extended;
On page 21, line 836: "w.r.t.";
On page 26, lines 927 and following: when you describe the two processing methods, I understood that the better performance of MWT could be due to the additional position correction external to the inverse algorithm. If I'm true, I suggest the authors to state even here the concept because from these lines it seems the result of a direct comparison of the two processing methods.
Reviewer 3 Report
Godd article with no language problems.
Despite this, the text would greatly benefit from including more figures to illustrate the use and/or results of the UAV/GPR systems, especially in the chapters 3 and 4.
